# Fathers' Parental Leave Uptake in Belgium and Sweden: Self-Evident or Subject to Employment Characteristics?

**Leen Marynissen [1,\*], Eleonora Mussino [2]** **, Jonas Wood [1] and Ann-Zofie Duvander [2]**

1   Department of Sociology, University of Antwerp, 2000 Antwerpen, Belgium; jonas.wood@uantwerp.be
2   Department of Sociology, University of Stockholm, 10691 Stockholm, Sweden;
    eleonora.mussino@sociology.su.se (E.M.); ann-zofie.duvander@sociology.su.se (A.-Z.D.)
\*   Correspondence: leen.marynissen@uantwerp.be

**Abstract:** The limited increase in fathers' involvement in childcare tasks in response to the unprecedented rise in female labour market participation illustrates the incomplete nature of the gender revolution. Available research provides evidence for micro-economic mechanisms and the influence of gender norms and social policy design on couples' gendered divisions of parental leave, but knowledge on how national level contexts shape partners' agency remains limited. Hence, comparative research from different national contexts is needed. This paper examines the association between fathers' pre-birth income and workplace characteristics, and whether they take up parental leave after the birth of their first child in Belgium and Sweden by using detailed longitudinal register data from Sweden and Belgium. Results show that, whereas an opportunity cost logic seems to underlie fathers' parental leave decisions in Belgium, gender equality in contributing to the household income yields the highest probability of fathers' parental leave uptake in Sweden. Furthermore, in Sweden, fathers' employment characteristics are more strongly associated with whether fathers' take leave longer than the quota than whether fathers take any leave at all. The different mechanisms in Belgium and Sweden suggest that the design of leave policies and the broader normative and institutional national level context moderate couples' parental leave uptake decisions.

**Keywords:** parental leave; fathers; Belgium; Sweden; employment characteristics

## 1. Introduction

In response to the unprecedented rise in women's educational and labour market participation and the associated rise in the share of dual-earner couples in recent decades, governments have increasingly developed social policies geared towards reconciling work and family life (Thévenon 2008). The so-called gender revolution is incomplete, however, as the increase in female labour force participation is not (yet) mirrored by an equivalent shift towards higher involvement of men in household work and child-rearing tasks (Goldscheider et al. 2015; McDonald 2000). The transition to parenthood, in particular, entails (an articulation of) traditional gender divisions of paid work (Dribe and Stanfors 2009; Kühhirt 2012; Neilson and Stanfors 2014), and the share of mothers taking up parental leave consistently exceeds the share of fathers taking up parental leave (Eydal et al. 2015; Geisler and Kreyenfeld 2011; RVA Dienst Studies 2014; Wood et al. 2018). This gender difference has motivated policy makers in several European countries to put forward higher male parental leave uptake as an explicit policy goal, whereas scholars have been enticed to unravel the mechanisms underlying gendered patterns of parental leave uptake. Several explanations have been put forward for the persistence of gender inequalities in couples' division of paid work around the transition to parenthood,

ranging from micro-economic and gendered moral rationalities on the individual or household level to parental leave policy designs and persistently gendered normative, institutional, or national contexts at the macro level. A limited, yet growing, body of literature documents these mechanisms and increasingly focuses on fathers' leave uptake in particular (Duvander 2014; Lappegård 2012; O'Brien and Wall 2017; Wood and Marynissen 2019). These studies identify partners' earnings (Lappegard 2008; Sundström and Duvander 2002), workplace characteristics (Bygren and Duvander 2006; Reich 2010), potentially gendered parenting norms and institutions (Wood and Marynissen 2019), and parental leave policy designs (Geisler and Kreyenfeld 2012; Mussino et al. 2018) as determinants of fathers' parental leave use.

Research on the determinants of fathers' parental leave use to date chiefly consists of single-country studies that have predominantly considered Nordic countries and Germany (Bygren and Duvander 2006; Duvander 2014; Duvander and Johansson 2012; Geisler and Kreyenfeld 2011; Haas et al. 2002; Lappegard 2008; Naz 2010; Reich 2010). These countries are all characterized by degenderizing parental leave policies—policies that promote the elimination of gender roles (Saxonberg 2013)[1]. Countries characterized by explicitly (or implicitly) genderizing parental leave policies on the other hand—i.e., most of continental West and Central European countries (Saxonberg 2013)—remain understudied (Lapuerta et al. 2011; Wood and Marynissen 2019). Furthermore, although previous research provides (some) evidence for micro-economic mechanisms (e.g., income loss) and the influence of gender norms and social policy design on couples' gendered divisions of parental leave, our knowledge on the extent to which national level contexts (e.g., family policies and overall level of gender equality) shape this behaviour (partners' agency) remains limited. Therefore, more comparative research is needed from different national contexts. Country contexts are complex where, for example, policies may have conflicting goals or different dimensions that may inhibit or facilitate fathers' parental leave use. To come closer to understand what policy dimensions are important, we need comparisons of both most similar and more different contexts.

This study compares to which extent fathers' parental leave uptake after the birth of the first child can be explained by their pre-birth (relative) labour market positions in Belgium and Sweden by using detailed longitudinal register data on dual earner couples. These two countries are quite similar with regard to the gender-equal setup of their parental leave systems (e.g., individual entitlement and equal length of leave for men and women) and also exhibit highly developed work–family reconciliation policies (e.g., widespread availability of formal childcare) (OECD 2016a; Population Council 2006). Although these contextual similarities may support similar determinants of fathers' leave uptake, both countries also differ in several dimensions which have been put forward as determinants of gender equality in leave-taking (Ciccia and Verloo 2012; Saxonberg 2013). The Swedish context exhibits generous income-related replacement benefits during parental leave, high flexibility, universal eligibility, and leave policies explicitly focusing on gender equality and fathers' involvement in childcare, such as the so-called "daddy quota". This contrasts with the Belgian laissez-faire leave setup with low flat-rate replacement benefits, limited flexibility, eligibility criteria connected to previous labour force participation, and the lack of specific policy measures to encourage fathers' participation in leave-taking. Hence, we expect that the association between fathers' pre-birth income from paid labour and workplace characteristics and their parental leave uptake differs between Belgium and Sweden, which may provide a possible explanation for the different levels of uptake in Belgium and Sweden and give an indication of national context contingencies.

The knowledge of couple characteristics associated with male parental leave uptake is of direct policy relevance in developed countries. Research has unveiled multiple positive effects of fathers' use of paternity and parental leave, such as higher gender equality with respect to the household

---

[1]　For Germany this only holds from the 2007 policy reform onwards. Before the reform, Germany was classified as implicitly genderizing (Saxonberg 2013).

division of childcare tasks and paid work in later years (Huerta et al. 2013; Patnaik 2018) as well as positive effects on children's development, fathers' personal development, family stability, and mothers' income and employment (Duvander and Jans 2009; Huerta et al. 2013). Hence, more insight in the determinants of fathers' parental leave could inform policies and generate multiple long-term positive effects. The Belgian context is hitherto rarely studied in the literature on the determinants of parental leave uptake. However, the comparison of determinants of parental leave in a context with low levels of leave use by fathers, such as Belgium, with those in a context with high levels of leave uptake by fathers, such as Sweden, could be of particular interest to policy makers in both Belgium and other continental West and Central European countries in view of promoting and supporting more gender equal parenting roles. The question for gender equality in parental leave is also placed high on the European policy agenda; higher uptake of family-related leave and flexible working arrangements by men is one of the main aims in the European Council's new directive on the work–life balance for parents and carers (EPRS 2019).

## 2. The Swedish and Belgian National Level Context

### 2.1. Parental Leave in Sweden

Most paid leave related to the birth of a child in Sweden is covered by parental leave, but Swedish social policy also distinguishes between temporary leave in connection with a child's birth or adoption (the former so-called 'daddy days' or paternity leave) apart from the actual parental leave. Maternity leave with benefit does not exist. However, mothers are obliged to take two weeks of mandatory absence from work before or after the birth of a child. The parental leave benefit can be used during this period (Duvander and Haas 2018). The temporary leave in connection with a child's birth or adoption allows the other parent (or carer) to take leave for a period of 10 days within the first 60 days after the birth of a child[2].

Parental leave, introduced in 1974, is residence based and covers all persons that are socially insured in Sweden and also those who are self-employed, unemployed, inactive, or enrolled in education. It is an individual partly-transferable right and consists of 240 days (approximately 8 months) of paid leave for the mother as well as 240 days of paid leave for the father. However, only 90 days are strictly reserved for the mother and 90 days are strictly reserved for the father[3]. Besides these quotas, the remaining days are transferable between parents. Parental leave can be taken up in days (similar to the uptake of regular vacation days) until the child reaches the age of 12 (8 years before 2014), with the restriction that most of the leave has to be used the first 4 years. Apart from full-time leave (i.e., a whole day), parents can also opt to take up leave in a part of a day, down to one-eighth of a day. In addition, as there is a right to leave (with or without pay) during the child's first 18 months, it is very common to mix paid and unpaid leave to extend the period at home. The parental leave benefit is income related for those who meet the criteria: For parents that have earnings above a certain threshold for at least 8 months before childbirth, 195 days are paid at 80% of previous earnings and 45 days are paid at a flat-rate benefit, presently at 18 euros a day (Duvander and Haas 2018). The 80% of prior earnings is often supplemented with another 10% by the employer through collective agreements. Parents who do not meet the criteria for an earning-related benefit receive a flat-rate benefit of approximately 25 euros a day (6 euros until 2002, then increased stepwise). Finally, the Swedish leave system provides certainty about future labour market positions as it entails job protection during leave (Duvander and Haas 2018).

---

[2]　During these ten days, the other parent (or carer) receives an income-related replacement benefit (77.6% of the prior earnings) paid by the Swedish Social Insurance Agency. All employees are eligible to take this leave.

[3]　The first quota month was introduced in 1995 and extended with a second month in 2002, and at the same time as the entire leave was extended with one month, a third month was added in 2016.

## 2.2. Parental Leave in Belgium

Belgian social policy more explicitly distinguishes between maternity leave, paternity leave, and parental leave than Swedish social policy. Maternity leave allows mothers to take leave for a minimum of 10 (obliged) and a maximum of 15 weeks. The replacement benefit during maternity leave is income-related[4]. Paternity leave allows fathers to take leave for a period of 10 days within the first four months after the birth of a child. These 10 days can be split over multiple periods. During the first three days, fathers' wages are paid by their employer. For the remaining 7 days, fathers receive an income-related benefit (82% of the gross wage) from the health insurance. Fathers who are inactive, unemployed, or self-employed are not entitled to paternity leave.

Apart from the maternity and paternity leaves, parental leave is an individual, non-transferable entitlement for both mothers and fathers. It was introduced in 1997. It allows each parent to take up full-time leave for a maximum of four months[5] at a flat-rate benefit (of 727 euros per month in 2010, the last observation year in our study) for each child younger than 12[6] (RVA Dienst Studies 2014). Full-time employees can also opt to reduce their working hours by 50% or 20% for a longer period[7], receiving a benefit that is reduced accordingly. The uptake of parental leave can be split over multiple time periods[8] depending on the sector of employment and previous work history, and periods of full-time and part-time leave can be combined (Desmet et al. 2007; RVA Dienst Studies 2014). An important feature of the Belgian parental leave system is that—in contrast to Sweden—only employees can make use of this policy, implying that the self-employed, unemployed, inactive as well as parents enrolled in education are excluded. Parents employed in the public sector or education sector are eligible without any conditions on working experience. Parents employed in the private sector, however, have to be working for their current employer for 12 out of 15 months prior to the application. This parental leave system provides certainty about future labour market positions, that is, certainty to return to full (previous) employment after a period of parental leave and protection against dismissal until three months following parental leave uptake. In addition, Belgian social policy includes other ways of temporary exits from work through a time credit system in the private sector and a similar scheme of career breaks in the public sector. These systems can be used to take care of a child younger than eight years old, but are not common in practice.

## 2.3. The Swedish and Belgian Broader National Level Context

Ciccia and Verloo (2012) have examined to what extent leave regulations in European countries promote gender equality in the family by using a fuzzy-set ideal type analysis. Their categorization of Sweden's and Belgium's parental leave policies in ideal types is exemplary for both countries' broader national contexts with respect to childcare and the division of paid and unpaid labour at the household level. Sweden is characterized as a limited caregiver model, entailing a high focus on gender equality, both in childcare—a responsibility initially placed primarily in the home—and in paid work, and a strong commitment to full employment. There are also strong incentives for parents to return to (full) employment within one year following the birth of a child, as long leaves are perceived to be

---

[4] 82% of the gross wage during the first 30 days of the maternity leave, 75% of the gross wage from the 31st day onwards. All mothers who are employed or unemployed at the moment of childbirth have a certain amount of work experience (≥120 days), are paid for social insurance, and are entitled to maternity leave (Rijksinstituut voor ziekte- en invaliditeitsverzekering 2019).

[5] Three months until 01.06.2012, four months from this date onwards, due to changes in regulation.

[6] From the introduction of parental leave in 1997, parents were entitled to leave only for children younger than four years. This age limit was raised to six years in 2005 and subsequently to 12 years in 2009.

[7] Eight months in case of a reduction of working hours with 50%, and 20 months in case of a reduction of working hours with 20%.

[8] In Belgium, parental leave can be split up in blocks of minimum 1 month of full-time leave, 2 months of part-time leave, or 5 months of 1/5th reduction of working hours. However, recent changes in legislation allow uptake of full-time leave in weeks and part-time leave in months if the employer agrees. It is thus a possibility, not a right. This legislation has been in force since 1 June 2019.

disadvantageous to both mothers' employment and children's development (Ciccia and Verloo 2012). Belgium is characterized as an unsupported universal breadwinner model, including full engagement of both men and women in the labour market. In this model, gender equality means gender sameness, implying that both partners should be freed from childcare responsibilities and thus a high degree of outsourcing family responsibilities to the state or market (e.g., subsidized childcare and service vouchers) (Ciccia and Verloo 2012).

Apart from parental leave regulations, both countries also differ in terms of parental employment patterns. With respect to the labour market and gender equality, Sweden does somewhat better than Belgium, but both countries still face challenges in this respect. Sweden exhibits high employment rates of both men and women (around 75%) and a low gender gap in employment (Eurofund 2016; OECD 2019). However, the gender gap in part-time employment is relatively high (around 25%) (Eurofund 2016). Also, both gender segregation in employment and the gender wage gap remain high in Sweden compared to other EU28 countries (European Institute for Gender Equality 2017a, 2017b; OECD 2019). Belgium, on the other hand, exhibits lower employment rates than Sweden for both men and women and a higher gender gap in both overall employment and part-time employment (Eurofund 2016). Although decreasing throughout the 2000s, female part-time employment is particularly high compared to most other EU28 countries (OECD 2019). Gender segregation in employment in Belgium is similar or somewhat lower than in Sweden (European Institute for Gender Equality 2017b). Belgium does explicitly well, better than Sweden, with regard to decreasing the gender pay gap (European Institute for Gender Equality 2017a; OECD 2019).

With respect to family policy and gender equality, Sweden is a leading country with all-encompassing gender policies, whereas Belgium exhibits more of a laissez-faire context. With its mild distinction between maternity, paternity, and parental leaves and the long well-paid leaves with high benefit ceilings and father quota, Swedish parental leave policies are called degenderizing (Saxonberg 2013). Accordingly, levels of parental leave uptake are high among both mothers and fathers: around 80% of the fathers and almost all mothers use parental leave in Sweden. Despite the gender equal setup that Belgian parental leave policies have in common with Sweden, this policy is called 'explicitly genderizing', meaning that it explicitly promotes different gender roles for men and women (Saxonberg 2013). This is mostly due to the long and well-paid maternity leave versus the short paternity leave, which introduces a gendered habitus among new parents and may therefore give rise to a gendered behaviour (Wood and Marynissen 2019). In addition, the low flat-rate replacement benefits, limited flexibility, and strict eligibility criteria provide few incentives for parents to take up parental leave. Also, in practice, access to parental leave remains subject to strong variation between employment sectors. As a result, despite a continuous increase throughout the early 2000s, parental leave uptake is relatively low among both fathers and mothers in Belgium[9]. In addition, the share of mothers taking up parental leave consistently exceeds the share of fathers taking up parental leave. This balance is however shifting: 92% of all parental leave that had been used in 2002, was used by mothers (RVA Dienst Studies 2014). This percentage decreased to 75% in 2010 (RVA Dienst Studies 2014). Also, the overwhelming majority of leave uptake is part-time or 1/5th, implying continued labour force participation. Hence, in a context in which parental leave uptake is not self-evident, leave uptake may entail higher or more penalties (e.g., forgone career opportunities), especially for fathers, compared to contexts where parental leave uptake is self-evident or even normative. Finally, both Sweden and Belgium have a long-standing history of subsidizing formal childcare services and exhibit the highest childcare coverage rates in Europe (Van Lancker and Ghysels 2012). As of the early 2000s, both countries are included in the list of countries that meet the Barcelona childcare targets of 33% enrolment for 0–2 year olds and 90% for children from three to six years of age (Population

---

[9] The National Public Employment Service (RVA) solely provides information on the number of persons taking up parental leave in each (month of a) year. Unfortunately, no information is available on the percentage of parents (fathers or mothers) that take up parental leave out of all (eligible) parents (Merla et al. 2018).

Council 2006). However, the use of formal childcare is skewed towards the higher incomes in Belgium, whereas there is no such pattern in Sweden and despite its high coverage rate, Flanders suffers from a shortage in childcare availability, which is, again, not the case for Sweden (Van Lancker and Ghysels 2012). On the other hand, while there is universal childcare availability in Sweden, public childcare is only available for children aged 1 or older, implying that uptake of parental leave or other informal care arrangements are necessary to care for children younger than 1 year of age. In Belgium, many mothers and most fathers only use their maternity and paternity leave and take up their parental leave part-time, later in the child's life, or not at all. Therefore, a large share of Belgian children starts to attend day care for at least some days a week when they are three months old.

## 3. Theoretical Perspectives

### 3.1. Income and Workplace Characteristics as Subject of Intra-Household Negotiation: Micro-Economic and Gendered Moral Rationalities

Fathers' parental leave uptake results from partners' joint decisions on their strategy to combine work and family. Hence, household-level mechanisms underlie father's parental leave use. Available literature suggests two different explanations for the gender division of paid work and household/childcare tasks between partners at the household level.

On the one hand, micro-economic theories imply that couples' parental leave uptake reflects partners' relative labour force potential, regardless of gender. According to Becker (1991) New Home Economics, partners pool their time and resources and develop work–family reconciliation strategies that maximize their joint utility. In contrast, bargaining theory and the relative resource perspective assert that parental leave uptake is the result of intra-household negotiations based on partners' resources, whereby both partners aim to maximize their individual utility (Blood and Wolfe 1960; Brines 1993). Although respecitvely joint versus individual utilities are maximized, the mechanisms in both theories are similar: taking up parental leave entails (at least some) income loss and may be perceived to entail possible career disadvantages and/or devaluation of human capital. As a result, the partner with the most favourable labour market position and opportunities is assumed to take up more paid work—and thus less parental leave—whereas the other partner will invest more time in household and childcare tasks, and will take up parental leave at a lower opportunity cost. Hence, in this view, fathers facing high opportunity costs in terms of income and/or career prospects will be less likely to take up parental leave than fathers facing lower costs in case of parental leave uptake (hypothesis 1). In terms of income, this would imply that fathers that contribute a higher share to the household income will be less likely to take up parental leave (hypothesis 1a).

Closely related to (perceived) career disadvantages are partners' workplace characteristics, which can act as gatekeepers to parental leave use, in particular for fathers. Although employees are entitled to take leave, organizations can either encourage or discourage their employees to take leave through informal means (e.g., stigmatization or career disadvantages, incite feelings of specialization and co-worker loyalty) (Bygren and Duvander 2006; Haas and Hwang 2019; Kaufman 2018; Moran and Koslowski 2019; van Breeschoten 2019). Particularly, organizations with a strong 'ideal worker' culture may discourage their employees to take leave (Haas and Hwang 2019). This 'ideal worker' culture is based on the male breadwinner model and entails that (male) employees should prioritize work above family responsibilities. Employees that take parental leave are considered to be less committed to their job and may thus face career disadvantages as a result of taking leave. Organizations with a large number of employees and a large share of women in their workforce are more likely to have routine practices in place when employees take up their parental leave (e.g., redistributing the work to other employees or hiring temporary replacements) (Haas and Hwang 2009). For these organizations, the cost of fathers' parental leave use is lower compared to organizations with fewer (female) employees. Hence, this gatekeeping mechanism based on workplace characteristics may result in higher leave uptake among fathers working in large (hypothesis 1b) and predominantly female organizations (hypothesis 1c) compared to fathers working in small and predominantly male organizations.

On the other hand, in addition or complementary to micro-economic rationality, persistent gendered parenting norms and collective understandings of what is best for men and women to do—so-called 'gendered moral rationalities'—influence couples' work–family combination and leave—strategies (Duncan and Edwards 1997). The 'doing gender' perspective argues that men and women conform to and reproduce these gender norms (Schneider 2011; West and Zimmerman 1987), as deviation from these gender norms could entail social penalties (e.g., stigmatization at the workplace). In this view, gendered uptake of parental leave reflects that mothers are still seen as the main caregivers, whereas fathers are expected to be the main breadwinner (Grunow and Evertsson 2016). As the previous paragraph indicates, this mechanism is also reflected in the normative aspect of workplaces' gatekeeping actions (e.g., male ideal worker). In sum, this explanatory mechanism suggests that persistent gendered parenting norms and preferences yield gendered uptake of parental leave (Brines 1993; Coltrane 2000; West and Zimmerman 1987), regardless of, or despite partners' pre-birth labour market resources. Although we are unable to operationalize and thus directly examine the effect of gender norms and gendered moral rationalities, this perspective should be kept in mind when interpreting the results.

*3.2. Institutional Factors*

Intra-household negotiations on couples' work–family combination strategies are not based solely on individual or household level factors. The design of leave policies and the broader normative and institutional national level context also shape the setting in which these decisions are made and thus shape fathers' agency (Pfau-Effinger 2004).

With respect to parental leave policies, several features have been found to significantly affect the level of uptake: (i) whether the income replacement benefit is sufficiently high, (ii) whether at least a part of the leave is strictly reserved for the father, and (iii) the flexibility of the leave (Ciccia and Verloo 2012; Geisler and Kreyenfeld 2012; Mussino et al. 2018; Patnaik 2018). These policy design features influence micro-economic considerations (income loss) as well as more normative considerations, e.g., the sense of entitlement to the leave as it is strictly reserved for the father. Well-compensated and individual-based leave (or daddy quotas) signals that parental leave is socially valued and makes fathers' care responsibilities visible at the workplace (Lammi-Taskula 2006; Närvi and Salmi 2019). The latter could in turn help fathers to overcome barriers to taking leave, such as perceived career disadvantages and social stigma (Patnaik 2018). Furthermore, high flexibility, such as the possibility to take up leave in days (full-time or part-time down to one-eighth of a day) in Sweden, may reduce the hurdle to take leave as it does not necessarily imply long-term absence from work. Limited flexibility, such as in Belgium where the parental leave can be split up in blocks of minimum 1 month of full-time leave, 2 months of part-time leave, or 5 months of 1/5th reduction of working hours, on the other hand, may increase the barrier for fathers to make use of their parental leave entitlement.

Next to parental leave policy design, the availability of more general social policies geared towards gender equality in both the labour market and childcare as well as the overall level of gender equality and the broader normative context shape the setting in which household-level parental leave uptake decisions are made. All-encompassing gender policies and a normative context in which childcare is considered a joint, instead of a predominantly female responsibility which is the case in Sweden (Ciccia and Verloo 2012), may encourage employers to facilitate the uptake of parental leave for both men and women. This, in combination with high flexibility and high income-related benefits, may overall lower the barrier for fathers to take up leave and therefore weaken the association between fathers' employment characteristics and their leave uptake. On the other hand, a laissez-faire policy and a more traditional normative context in which childcare is still perceived to be primarily the mother's responsibility, which is the case in Belgium, may consolidate or at least not counter (male) ideal worker cultures at the organizational level. This, in combination with limited flexibility and low flat-rate benefits, may consolidate or even reinforce the barriers to fathers' leave-taking. Therefore, fathers' income and workplace characteristics may be stronger determinants of their leave uptake in these contexts compared to more supportive contexts. Hence, we expect different associations between

fathers' share of the household income and their workplace characteristics, and fathers' parental leave uptake for Belgium compared to Sweden (hypothesis 2).

In addition, for Sweden in particular, we expect stronger associations between fathers' share of the household income and their workplace characteristics, and fathers' leave use exceeding the quota than between fathers' employment characteristics and their leave use overall (hypothesis 3). Although the overall barriers to take parental leave may be lower in Sweden, we expect that using more than the father's quota may be much more subject to intra-household negotiations based on fathers' income and workplace characteristics for two reasons: (i) whereas not using the quota implies renouncing a benefit as it cannot be transferred to the mother, using more than the quota implies using a part of the leave that is transferable and (ii) using more than the quota implies longer absence from work and thus possibly higher costs in terms of income and career disadvantages. Hence, the strength and direction of the association between fathers' employment characteristics and their leave uptake of more than the quota in Sweden may be more similar to the Belgian case than is the strength and direction of the association between fathers' employment characteristics and their leave uptake overall in Sweden.

## 4. Data and Methods

### 4.1. Data

We used data from the Belgian Administrative Socio-Demographic Panel (BASD Panel) that was constructed using microdata from the National Register and the Crossroads Bank for Social Security. This panel provides detailed longitudinal information on a representative sample of 108,511 women aged 15–50 years, legally residing in Belgium in the period from 1 January 1999 to 31 December 2010. To maintain the cross-sectional representativeness of the panel throughout the observation period, annual top-up samples of 15 year olds were drawn as well as annual samples of women aged 16–50 years who settled in Belgium in the preceding year. In addition to the sampled individuals, the panel includes all household members who, in any year of the observation period, resided in the households of sampled women on the 1st of January. As a result, the panel provides a representative sample of heterosexual, cohabiting couples. The BASD Panel also provides detailed quarterly information on labour market positions and income of all household members and annual information on household composition. An important note with respect to the Belgian data is that it does not allow to distinguish between parental leave and other types of leave embedded in the Belgian 'time credit' system (Merla et al. 2018). Hence, the uptake of leave measured in this study refers to all types of Belgian 'time credit' leaves. Previous research, however, demonstrates that most leave-taking immediately following childbirth is parental leave (Desmet et al. 2007; Kil et al. 2018). For Sweden, we used data from the population registers, which cover the total population living in Sweden (STAR—Sweden over Time: Activities and Relations). This database contains information on the dates of all demographic events and annual information on socio-demographic variables, such as education, income, labour market attachment, and parental leave benefit days.

We observed couples who had their first child between 2001 and 2007, who were cohabiting for at least one year before the birth of the child, and where both partners had an income from paid labour one year before the birth. The Swedish data is annual, so to observe couples' parental leave use for exactly two years after the birth of a child, we only selected couples who had their child in December. For Belgium, we included couples who had their first child in any month of the year. The quarterly measurements were aggregated to yearly observations for several years before and after the birth of a child to conform to the Swedish data. We observed these couples for two years following the birth of their first child. The Belgian analytic sample provided information on 3480 couples. The Swedish analytic sample consisted of 11,152 couples[10].

---

[10] Given that the Belgian sample stems from large, yet sampled panel data whereas the Swedish sample stems from full register data, the Belgian analytical sample is considerably smaller than the Swedish analytical sample. As a result, even

*4.2. Variables*

The dependent variable was a dummy variable with a value one if the father used parental leave within the two years after the birth of the first child and a value of zero if the father did not use any leave. Because we expected that fathers' leave uptake of more than the quota may be much more subject to intra-household negotiations based on fathers' income and workplace characteristics, we considered an additional outcome variable for Sweden indicating whether father's leave uptake exceeded the quota, which will henceforth be referred to as leave+. Because the quota increased during the observation period to two months in 2002, the leave+ outcome variable referred to more than 30 days of use for children born until 2001, and more than 60 days for children born after that date. Table 1 provides the distribution of both outcome variables as well as the distribution of all covariates.

**Table 1.** Summary statistics.

|  | Sweden | | Belgium | |
|---|---|---|---|---|
|  | **Freq.** | **Perc.** | **Freq.** | **Perc.** |
| **Fathers' leave uptake** | | | | |
| 0 | 1310 | 11.75 | 3279 | 94.22 |
| 1 | 9842 | 88.25 | 201 | 5.78 |
| **Fathers' uptake of leave+** | | | | |
| 0 | 5741 | 51.48 | | |
| 1 | 5411 | 48.52 | | |
| **Household income** | | | | |
| First tertile | 3676 | 32.96 | 1188 | 34.14 |
| Second tertile | 3739 | 33.53 | 1216 | 34.94 |
| Third tertile | 3737 | 33.51 | 1076 | 30.92 |
| **Father's share household income** | | | | |
| father >75% | 386 | 3.46 | 195 | 5.6 |
| father 55%–75% | 5414 | 48.55 | 1629 | 46.81 |
| father 45%–55% | 4149 | 37.2 | 1237 | 35.55 |
| father <45% | 1203 | 10.79 | 419 | 12.04 |
| **Workplace size** | | | | |
| <5 | 125 | 1.12 | 218 | 6.26 |
| 5–9 | 196 | 1.76 | 209 | 6.01 |
| 10–19 | 343 | 3.08 | 315 | 9.05 |
| 20–49 | 759 | 6.81 | 434 | 12.47 |
| 50–99 | 831 | 7.45 | 319 | 9.17 |
| 100–199 | 729 | 6.54 | 313 | 8.99 |
| 200–499 | 866 | 7.77 | 373 | 10.72 |
| 500–999 | 423 | 3.79 | 282 | 8.1 |
| 1000< | 1288 | 11.55 | 1017 | 29.22 |
| Unknown | 5592 | 50.14 | | |
| **Share women workplace/sector** | | | | |
| 0%–30% | 2534 | 22.72 | 1527 | 43.88 |
| 30%–70% | 2278 | 20.43 | 1599 | 45.95 |
| 70%–100% | 748 | 6.71 | 353 | 10.14 |
| Unknown | 5592 | 50.14 | 1 | 0.03 |
| **Origin** | | | | |
| Both Swedish/Belgian origin | 9542 | 85.56 | 2552 | 73.33 |
| Both migrant origin | 427 | 3.83 | 259 | 7.44 |
| Father migrant origin | 506 | 4.54 | 353 | 10.14 |
| Mother migrant origin | 677 | 6.07 | 289 | 8.3 |
| Unknown | | | 27 | 0.78 |

moderate effects in the Swedish analyses will be more likely to reach statistical significance. Therefore, we do not put too much emphasis on the statistical significance of our results, but rather focus on the patterns of effects.

**Table 1.** *Cont.*

|  | Sweden | | Belgium | |
|---|---|---|---|---|
|  | **Freq.** | **Perc.** | **Freq.** | **Perc.** |
| **Age father** | | | | |
| ≤25 | 1083 | 9.71 | 175 | 5.03 |
| 26/30 | 3914 | 35.1 | 1431 | 41.12 |
| 31/35 | 3954 | 35.46 | 1293 | 37.16 |
| 36/40 | 1556 | 13.95 | 420 | 12.07 |
| ≥41 | 645 | 5.78 | 161 | 4.63 |
| **Age mother** | | | | |
| ≤25 | 2093 | 18.77 | 529 | 15.2 |
| 26/30 | 4875 | 43.71 | 1849 | 53.13 |
| 31/35 | 3233 | 28.99 | 899 | 25.83 |
| 36/40 | 829 | 7.43 | 186 | 5.34 |
| ≥41 | 122 | 1.09 | 17 | 0.49 |
| **Father's education** | | | | |
| Primary | 774 | 6.94 | 176 | 5.06 |
| Secondary | 5458 | 48.94 | 332 | 9.54 |
| Tertiary | 4920 | 44.12 | 410 | 11.78 |
| Unknown | | | 2562 | 73.62 |
| **Mother's education** | | | | |
| Primary | 512 | 4.59 | 169 | 4.86 |
| Secondary | 4537 | 40.68 | 684 | 19.66 |
| Tertiary | 6103 | 54.73 | 1101 | 31.64 |
| Unknown | | | 1526 | 43.85 |
| **Birth year of child** | | | | |
| 2001 | 1370 | 12.28 | 469 | 13.48 |
| 2002 | 1508 | 13.52 | 446 | 12.82 |
| 2003 | 1553 | 13.93 | 450 | 12.93 |
| 2004 | 1632 | 14.63 | 498 | 14.31 |
| 2005 | 1739 | 15.59 | 561 | 16.12 |
| 2006 | 1620 | 14.53 | 514 | 14.77 |
| 2007 | 1730 | 15.51 | 542 | 15.57 |
| **Second child within X years** | | | | |
| 0 (no child within 2 years) | 8785 | 78.78 | 2612 | 75.06 |
| First year | 23 | 0.21 | 106 | 3.05 |
| Second year | 2344 | 21.02 | 762 | 21.9 |
| **Civil status** | | | | |
| Unmarried cohabiting | 6355 | 56.99 | 1311 | 37.67 |
| Married | 4792 | 42.97 | 2169 | 62.33 |
| Unknown | 5 | 0.04 | | |
| **NUTS1** | | | | |
| East Sweden/Flanders(BE) | 4782 | 42.88 | 2321 | 66.7 |
| South Sweden/Wallonia(BE) | 4735 | 42.46 | 905 | 26.01 |
| North Sweden/Brussels(BE) | 1635 | 14.66 | 254 | 7.3 |
| Total | 11,152 | 100 | 3480 | 100 |

Source: BASD Panel 1999–2010 and STAR, calculations by authors.

One of the main independent variables of interest was the father's share of the household income one year before the birth of the first child. *Household income* is the sum of both partners' incomes from paid labour and it was divided into tertiles. *Father's share of the household income* was categorized into four categories in our models: (i) male breadwinners, where the father earned more than 75% of the household income, (ii) "1.5 earners", where the father earned between 55% and 75% of the household income, (iii) dual earners, where both partners had similar incomes and the father earned between 45% and 55% of the household income, and (iv) female breadwinners, where the father earned less than 45% of the household income.

The other two main independent variables of interest were workplace size and share of women in the workplace, again measured one year before birth. For Sweden, these variables came from the Swedish occupational register that only includes a sample of private firms with less than 500 employees. The sampling strategy is related to the size of the firm; larger firms are more likely to be sampled than smaller firms. Previous studies have done sensitivity analyses to confirm that the sample is representative of the entire population of workers in small firms (Ohlsson-Wijk 2015). Although the Swedish sample size of the analyses with workplace characteristics was smaller due to the sampling of workplace information, the smaller sample was not a selective group. *Workplace size* is a predefined categorical variable representing the number of employees at the workplace, ranging from less than five employees to 1000 or more employees (values: <5, 5–9, 10–19, 20–49, 50–99, 100–199, 200–499, 500–999, ≥1000). *Share of women* in the workplace/employment sector is the percentage of women working in the father's workplace/employment sector, divided into three categories: (i) 0–30%, (ii) 30–70%, and (iii) 70–100%. For Sweden, this variable captured the share of women in the workplace. For Belgium, this indicator was constructed using the panel data at hand. In contrast to the Swedish register data, the BASD Panel does not cover all individuals working in a workplace. Hence, to achieve a more solid estimation, we did not calculate the share of women at the workplace, but the share of women in the employment sector.

Further, we controlled for a number of socio-demographic characteristics known to be associated with the probability of fathers' use of parental leave. We added a linear and squared term for the father's age at childbirth and a categorical indicator of age of the mother at childbirth (with values: ≤25, 26/30, 31/35, 26/40, ≥41) and controlled for the birth of a second child, whether the couple was married or cohabiting, and level of education of both parents. As previous research indicates that parental leave uptake varies by origin (Kil et al. 2018; Mussino and Duvander 2016), we controlled for couples' origin, distinguishing couples where (i) both partners had a Swedish/Belgian origin, (ii) both partners had a migrant origin, (iii) the father had a migrant origin, or (iv) the mother had a migrant origin. We also controlled for region of residence (Nuts1) because, at least in Belgium-family policies, policy coverage and labour market conditions differ between regions. In Sweden, the policy is national, but the take-up may vary between areas. Lastly, we controlled for the year of birth of the child as available research indicated an increase in fathers' parental leave use throughout the early 2000s.

*4.3. Analyses*

We used linear probability models (LPMs) to estimate the probability of using parental leave among fathers in the two countries in a stepwise approach. We preferred LPMs rather than logit models, because the latter have been shown to suffer from omitted variable bias (Mood 2009)[11]. Model I included all control variables and the income variables. Model II additionally included workplace/sector variables. For Sweden, the selection of couples included in model II was smaller than in model I as we did not have information on workplace variables for all fathers in the initial sample. Furthermore, for Sweden, these models were run for the two different outcome variables.

## 5. Results

*5.1. Descriptive Results*

Between 2001 and 2010, only 5.8% of the Belgian fathers in the selected sample used parental leave in the first two years following the birth of their first child. This contrasts with Sweden, where during the same period 88.3% of the fathers used parental leave and 48.5% of the fathers exhibited uptake above and beyond the quota (Table 1). Figure 1 shows the percentage of fathers using parental leave in

---

[11] As a result, the interpretation of odds ratios becomes problematic as they also reflect unobserved heterogeneity, which also makes it problematic to compare odds ratios across models with different independent variables and across samples—the main focus here (Mood 2009).

Sweden and Belgium by year of birth of the first child. In Sweden, fathers' parental leave use exceeded 80% in all observation years. The use of leave+ dropped below 40% for fathers with children born in 2002 and then slowly increased again, exceeding 50% in 2007. This drop could be accounted for by the change in Swedish leave regulations, extending the father's quota from one to two months for children born in 2002 or later. In Belgium, fathers' overall parental leave use was low compared to the Swedish levels. Nevertheless, a steady increase in fathers' leave use was observed between 2001 and 2007.

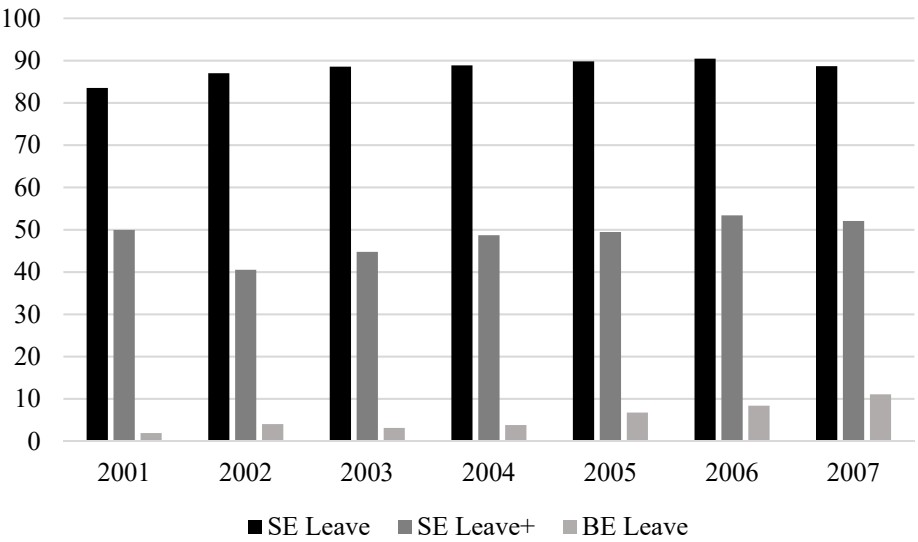

**Figure 1.** Fathers' leave use by birth year of the first child.

### 5.2. Multivariate Results

Table 2 shows the results of the linear probability models of fathers' parental leave use in Sweden, with separate models for (i) fathers using leave (overall) and (ii) fathers using leave+. Table 3 shows the results of the linear probability models of fathers' uptake of parental leave in Belgium.

With respect to fathers' share of the household income, fathers in equal earner households (where the father earns between 45% and 55% of the household income) had the highest probability of taking leave and leave+ in Sweden. Thus, the micro-economic logic of minimizing opportunity costs (hypothesis 1a) did not seem to prevail in Sweden. When controlling for workplace characteristics (Model IIa, Table 2), the probability of fathers' leave and leave+ in female breadwinner households did not differ from the probability of fathers' leave and leave+ in equal earner households. The results for Belgium, though not all significant, showed a clear opportunity cost logic: the higher the fathers' share in the household income, the lower the probability of using parental leave. These results confirmed hypothesis 1a for Belgium. As we will elaborate on later, we considered the low and flat-rate benefits as the main explanation for this finding; the larger the father's share of the household income, the larger the (household's) income loss in case of leave uptake by the father.

**Table 2.** Linear probability models of fathers' parental leave use in Sweden.

| | (a) Father Uses Leave vs. No Leave Use | | | | (b) Father Uses More Than Quota vs. No or Less Leave | | | |
| --- | --- | --- | --- | --- | --- | --- | --- | --- |
| | **Model Ia** | | **Model IIa** | | **Model Ib** | | **Model IIb** | |
| | *Coeff.* | *Sig.* | *Coeff.* | *Sig.* | *Coeff.* | *Sig.* | *Coeff.* | *Sig.* |
| **Constant** | 0.750 | *** | 0.820 | *** | 0.141 | | −0.075 | |
| **Age father** | 0.006 | | 0.003 | | 0.023 | ** | 0.038 | *** |
| **Age father, squared** | 0.000 | | 0.000 | | 0.000 | *** | −0.001 | *** |
| **Age mother (ref. 31/35 years)** | | | | | | | | |
| ≤25 | 0.007 | | −0.001 | | −0.007 | | −0.005 | |
| 26/30 | 0.011 | | 0.009 | | −0.005 | | −0.005 | |
| 36/40 | 0.010 | | 0.023 | | −0.017 | | −0.024 | |
| ≥41 | −0.053 | | −0.004 | | −0.050 | | −0.030 | |
| **Education father (ref. Primary)** | | | | | | | | |
| Secondary | 0.043 | ** | −0.016 | | 0.045 | * | 0.005 | |
| Tertiary | 0.037 | ** | −0.030 | | 0.136 | *** | 0.059 | |
| **Education mother (ref. Primary)** | | | | | | | | |
| Secondary | 0.007 | | 0.028 | | −0.033 | | −0.043 | |
| Tertiary | 0.031 | * | 0.056 | * | 0.112 | *** | 0.113 | ** |
| **Origin (ref. both Swedish)** | | | | | | | | |
| Both migrant origin | −0.144 | *** | −0.135 | *** | −0.215 | *** | −0.220 | *** |
| Father migrant origin | −0.062 | *** | −0.059 | *** | −0.052 | ** | −0.048 | |
| Mother migrant origin | −0.080 | *** | −0.081 | *** | −0.112 | *** | −0.098 | ** |
| **Married (ref. Unmarried cohabiting)** | | | | | | | | |
| Married | −0.003 | | 0.003 | | 0.037 | *** | 0.044 | ** |
| Unknown | 0.186 | | 0.131 | | −0.078 | | −0.418 | |
| **Nuts1 (ref. East Sweden)** | | | | | | | | |
| South Sweden | −0.008 | | −0.015 | | −0.106 | *** | −0.123 | *** |
| North Sweden | −0.017 | | −0.012 | | −0.042 | ** | −0.042 | * |
| **Birth year child (ref. 2001)** | | | | | | | | |
| 2002 | 0.034 | ** | 0.045 | ** | −0.094 | *** | −0.126 | *** |
| 2003 | 0.046 | *** | 0.039 | * | −0.068 | *** | −0.091 | *** |
| 2004 | 0.048 | *** | 0.056 | *** | −0.033 | | −0.052 | * |
| 2005 | 0.059 | *** | 0.056 | *** | −0.020 | | −0.028 | |
| 2006 | 0.066 | *** | 0.044 | ** | 0.012 | | −0.010 | |
| 2007 | 0.050 | *** | 0.064 | *** | 0.002 | | −0.007 | |
| **Second child within X years (ref. No 2nd child within 2 years)** | | | | | | | | |
| First year | 0.036 | | 0.088 | | 0.059 | | −0.055 | |
| Second year | −0.005 | | −0.014 | | −0.014 | | −0.028 | |
| **Household income (ref. Second tertile)** | | | | | | | | |
| First tertile | −0.022 | ** | −0.021 | * | −0.009 | | −0.007 | |
| Third tertile | −0.013 | | −0.012 | | 0.027 | * | 0.040 | * |
| **Father's share household income (ref. similar income (45%/54%))** | | | | | | | | |
| Male breadwinner (≥75%) | −0.089 | *** | −0.096 | *** | −0.148 | *** | −0.144 | *** |
| 1.5 earners (≥55%/<75%) | −0.028 | *** | −0.042 | *** | −0.083 | *** | −0.095 | *** |
| Female breadwinner (<45%) | −0.049 | *** | −0.016 | | −0.034 | * | −0.013 | |
| **Workplace size (ref. < 5)** | | | | | | | | |
| 5–9 | | | 0.013 | | | | 0.079 | |
| 10–19 | | | 0.033 | | | | 0.074 | |
| 20–49 | | | 0.042 | | | | 0.089 | |
| 50–99 | | | 0.020 | | | | 0.078 | |
| 100–199 | | | 0.050 | | | | 0.121 | ** |
| 200–499 | | | 0.048 | | | | 0.120 | ** |
| 500–999 | | | 0.023 | | | | 0.154 | ** |
| ≥1000 | | | 0.046 | | | | 0.123 | ** |
| **Share women workplace (ref. 30%–70%)** | | | | | | | | |
| 0%–30% | | | 0.011 | | | | −0.007 | |
| 70%–100% | | | 0.003 | | | | 0.046 | * |
| N couples | 11,152 | | 5560 | | 11,152 | | 5560 | |

Significance levels: * *p* < 0.05; ** *p* < 0.01; *** *p* < 0.001; Source: STAR, calculations by authors.

**Table 3.** Linear probability models of fathers' parental leave use in Belgium.

| | Father Uses Leave vs. No Leave Use | | | |
| --- | --- | --- | --- | --- |
| | Model I | | Model II | |
| | *Coeff.* | *Sig.* | *Coeff.* | *Sig.* |
| **Constant** | 0.186 | | 0.114 | |
| **Age father** | −0.010 | | −0.010 | |
| **Age father, squared** | 0.000 | | 0.000 | |
| **Age mother (ref. 31/35 years)** | | | | |
| ≤25 | 0.023 | | 0.020 | |
| 26/30 | 0.004 | | 0.002 | |
| 36/40 | 0.001 | | 0.001 | |
| ≥41 | −0.091 | | −0.078 | |
| **Education father (ref. Primary)** | | | | |
| Secondary | −0.002 | | 0.002 | |
| Tertiary | 0.006 | | 0.008 | |
| Unknown | 0.007 | | 0.008 | |
| **Education mother (ref. Primary)** | | | | |
| Secondary | −0.017 | | −0.018 | |
| Tertiary | −0.012 | | −0.013 | |
| Unknown | −0.009 | | −0.010 | |
| **Origin (ref. both Belgian)** | | | | |
| Both migrant origin | 0.022 | | 0.020 | |
| Father migrant origin | 0.011 | | 0.012 | |
| Mother migrant origin | −0.009 | | −0.009 | |
| Unknown | 0.045 | | 0.045 | |
| **Married (ref. Unmarried cohabiting)** | −0.011 | | −0.011 | |
| **Nuts1 (ref. Flanders)** | | | | |
| Wallonia | −0.032 | ** | −0.032 | ** |
| Brussels | −0.013 | | −0.012 | |
| **Birth year of child (ref. 2001)** | | | | |
| 2002 | 0.017 | | 0.017 | |
| 2003 | 0.002 | | 0.003 | |
| 2004 | 0.028 | | 0.011 | |
| 2005 | 0.055 | * | 0.043 | ** |
| 2006 | 0.048 | ** | 0.058 | *** |
| 2007 | 0.093 | *** | 0.086 | *** |
| **Second child within X years (ref. No 2nd child within 2 years)** | | | | |
| First year | 0.035 | | 0.036 | |
| Second year | 0.008 | | 0.007 | |
| **Household income (ref. Second tertile)** | | | | |
| First tertile | −0.041 | *** | −0.034 | ** |
| Third tertile | −0.030 | ** | −0.031 | ** |
| **Father's share household income (ref. similar income (45%/54%))** | | | | |
| Male breadwinner (≥75%) | −0.031 | | −0.037 | * |
| 1.5 earners (≥55%/<75%) | 0.000 | | −0.003 | |
| Female breadwinner (<45%) | 0.018 | | 0.022 | |
| **Workplace size (ref. < 5)** | | | | |
| 5–9 | | | 0.045 | * |
| 10–19 | | | 0.032 | |
| 20–49 | | | 0.033 | |
| 50–99 | | | 0.047 | * |
| 100–199 | | | 0.049 | * |
| 200–499 | | | 0.080 | *** |
| 500–999 | | | 0.082 | *** |
| ≥1000 | | | 0.062 | *** |
| **Share women workplace (ref. 30%–70%)** | | | | |
| 0%–30% | | | 0.023 | ** |
| 70%–100% | | | 0.017 | |
| N couples | 3480 | | 3480 | |

Significance levels: * $p < 0.05$; ** $p < 0.01$; *** $p < 0.001$; Source: BASD Panel 1999–2010, calculations by authors.

Whereas the probability of fathers' parental leave use did not vary by workplace size in Sweden, the probability of fathers' parental leave+ use did slightly increase by workplace size. Fathers working at workplaces with 100 employees or more had a higher probability of taking leave+ compared to fathers working at workplaces with less than five employees. Similar associations were found for Belgium. The larger the workplace size, the larger the probability that the father used parental leave. These results thus confirmed hypothesis 1b for Belgium and Sweden with respect to leave+. The association with the share of women in the workplace in Sweden followed a similar pattern. There was a minor positive association between working in a female-dominated workplace (>70% female employees) and fathers using leave+ in Sweden, when compared with a gender balanced workplace. These findings were in line with our expectations based on micro-economic and gendered moral rationalities (hypothesis 1c). No significant associations were found with respect to fathers' overall leave use in Sweden. For Belgium, against our expectations (hypothesis 1c), the results showed a small positive association between working in a male-dominated sector (<30% female employees) and parental leave uptake, when compared with working in a more gender-balanced sector. However, the category of male-dominated employment sectors comprised only two sectors in which 44% of the fathers in our sample worked in: (i) agriculture, mineral extraction, or industry and (ii) logistics, storage, or distribution. These sectors, however, exhibited only slightly higher levels of parental leave uptake by fathers (7% and 6%, respectively) compared with gender-balanced employment sectors (with uptake levels on average of 4 to 5%). Moreover, the category of female-dominated sectors comprised only three sectors in which 12% of the fathers in our sample worked in: (i) education, (ii) health services or social care, and (iii) art, leisure, recreation, or other services (from 2006 onwards). Whereas the latter two sectors in fact did exhibit higher levels of parental leave uptake by fathers (11% and 14%, respectively), the education sector exhibited very low levels of parental leave uptake by fathers (3.8%). Education is a sector in which full-time working hours are already relatively compatible with family life and children's school attendance, presumably lowering the need for parental leave uptake.

We further controlled for household income and a number of socio-demographic characteristics known to be associated with the probability of fathers' use of parental leave. For Sweden, a household income in the lowest tertile negatively affected fathers' leave uptake in Sweden, while there was a positive association between a household income in the highest tertile and fathers' leave+ compared to having a household income in the second tertile. Hence, the association with household income differed somewhat between models where we looked at whether the father used leave at all and whether he used a long leave. Furthermore, the results on fathers' leave use for Sweden (Table 2, models I and IIa) showed no variation by the age of the parents, the couples' marital status, their region of residence, and whether a second child was born within the two years following the first birth. There was, however, a clear association between both partners' origin and the fathers' parental leave uptake; whenever one of both partners had a migrant origin, the probability of leave uptake by the father was lower compared to when both partners had a Swedish origin. In addition, there was a steady positive association between the birth year of the first child and fathers' parental leave uptake, representing a period trend. Fathers with a child born in more recent years had higher probabilities of taking parental leave than fathers with a child born in 2001. With regard to level of education, fathers with secondary or tertiary education had higher probabilities of using leave than fathers with only primary education. However, this association turned insignificant when controlled for workplace characteristics, which also meant reducing the sample size. In contrast to the results on fathers' leave use, the results on fathers' leave+ use (Table 2, models I and IIb) showed a positive association between father's age (linear) with their parental leave uptake and a significant but close to zero association with the squared term, indicating that the probability of taking up leave+ increased with fathers' age. Furthermore, whereas the association between fathers' education and their parental leave uptake was similar for leave and leave+ uptake, mothers' level of education was more strongly and positively associated with fathers' leave+ use than fathers' leave use. In addition, married fathers and fathers with a Swedish origin had a higher probability of using leave+ than cohabiting fathers and fathers

with a migrant origin, and fathers living in South or North Sweden had a lower probability of using leave+ than fathers living in East Sweden (which includes the Stockholm area). Finally, the positive trend in fathers' leave uptake by the birth year of the child was not mirrored in the analyses of leave+ (Table 2, models Ib and IIb). The probability of fathers' leave uptake was lower when the child was born in 2002 and 2003 compared to when the child was born in 2001. The probability of taking leave+ for children born between 2004 and 2007 did not significantly differ from the probability of taking leave+ for a child born in 2001. As suggested in the descriptive results, this was probably due to the extension of the quota from one to two months in 2002.

For Belgium, fathers in couples with a household income in the second tertile had a higher probability of leave uptake than fathers in couples with a household income in the first and third tertile. The lower probability of using leave among fathers in the third tertile was in line with the larger opportunity costs in case of a higher income. The lower probability of using leave by fathers in the first tertile income category could also be explained by micro-economic opportunity cost logic: although the absolute amount of income loss was smaller in low-income households compared to middle- or high-income households, the income loss might be felt more in these low-income households as they were likely to have less economic margin. Furthermore, fathers' parental leave uptake in Belgium did not significantly vary by the age of the mother and the father, either partners' educational level, the couples' migration background, marital status, or whether a second child was born within the two years following the first birth. There was variation, however, in father's parental leave use between regions and over time, with a significantly lower probability of father using leave in Wallonia compared to Flanders, and higher uptake probabilities for fathers with a child born in 2005 or later (Table 3).

## 6. Discussion and Conclusions

Despite scholars' and policy makers' increasing interest in the involvement of fathers and gender equality at the household level, our knowledge on the association between fathers' labour force positions and their parental leave uptake in different normative and institutional national contexts remains limited to predominantly Nordic countries. Using longitudinal couple data from Swedish population registers and the Belgian Administrative Socio-Demographic Panel, this study examines the association between fathers' pre-birth relative income and workplace characteristics and their parental leave uptake after the transition to parenthood in Sweden and Belgium. We compare the micro level link between fathers' employment characteristics and their parental leave uptake between countries with a similar focus on high labour market participation of both fathers and mothers, but also substantial differences with respect to gender equality on the labour market, in social policy and with respect to childcare.

Based on our results, this study reaches two conclusions. First, the association between fathers' pre-birth employment characteristics and their parental leave uptake differs between Belgium and Sweden, confirming hypothesis 2. These differential associations are discussed in the light of crucial differences between both countries with respect to the design features of parental leave as well as the broader institutional and gender context these policies are embedded into. Second, in Sweden, fathers' income and workplace characteristics are more strongly associated with whether fathers take leave longer than the quota than whether fathers take any leave at all, confirming hypothesis 3. Hence, the findings of this comparative study suggest that the design of leave policies and the broader normative and institutional national level contexts moderate couples' parental leave uptake strategies and that, although being a forerunner country in family policies and gender equality, Sweden also still faces challenges with respect to fathers' leave uptake for a longer period of time.

With respect to income, fathers most often use leave when there is gender equality in contributions to the household income in Sweden, which is in line with findings for Norway (Lappegard 2008). In Belgium, the micro-economic logic of minimizing fathers' and households' opportunity costs seems to prevail in parental leave decisions, which is similar to patterns found in Germany (before the reform), the Netherlands, and the UK (Geisler and Kreyenfeld 2011; Kaufman 2018; Reich 2010; van

Breeschoten 2019). We consider the low- and flat-rate benefit in Belgium versus the high income-related benefit during parental leave in Sweden as a crucial explanation for the different association between relative income and fathers' leave use in Belgium and Sweden. First of all, the income loss when using parental leave is substantial in Belgium as the benefit is far below the minimum wage. Furthermore, since it is a flat-rate benefit, the income loss quickly becomes disproportionally large once individuals or households move up the income ladder. The larger the father's share in the household income, the higher is the income loss in case the father takes up parental leave. In addition, the limited flexibility in comparison to the Swedish context—with uptake in blocks of one month full-time, two months part-time, or five months of 1/5th leave—implies a large income loss for a considerable period. The large opportunity costs, in combination with Belgium's laissez-faire policy with respect to gender and parental leave and its unsupported universal breadwinner model that focuses on outsourcing instead of family responsibilities, result in very low levels of parental leave uptake by fathers, particularly by main breadwinner fathers. In Sweden, on the other hand, the benefit is income related up to a certain (relatively high) ceiling. So although also in Sweden, the absolute income loss is larger for individuals or households at the higher end of the income distribution, the loss is proportionally larger—i.e., always 20% of the income. In addition, parental leave can be taken up in days or even part of a day, meaning that the income loss can be spread out over longer or more time periods. Hence, the high focus on gender equality in both childcare and paid work—that is inherent to the limited caregiver model—seems to prevail in fathers' parental leave uptake decisions in Sweden rather than an opportunity cost logic. Furthermore, the lack of significant associations between workplace characteristics and fathers' leave use in Sweden seems to suggest that the use of some parental leave, up to the quota, has become a quasi-universal and accepted practice. Using more than the quota, however, seems to be more subject to micro-economic considerations and institutional constraints (such as workplaces acting as gatekeepers)—although even then fathers' uptake of leave longer than the quota is relatively high, certainly compared to Belgium.

We identify two limitations of this study and corresponding avenues for future research. First, it is important to note that cautiousness is needed with respect to the causal interpretation of this study's results. Income and workplace characteristics may affect fathers' leave uptake, but also mechanisms of self-selection may be at play. More family-oriented individuals may self-select into labour market positions that enable the harmonization of work and family life. Hence, these individuals may move into employment sectors with high levels of flexibility or higher wages in view of parenthood. Therefore, future work using a longer observation window may consider looking into the effect of partners' employment histories—explicitly taking into account (recent) job moves or changes in working hours or income—on parental leave uptake and potentially shed light on self-selection mechanisms. Second, although the datasets used in this study entail detailed longitudinal microdata on couples and their employment characteristics, they each have their limitations with respect to data availability. For Belgium, the coverage of educational information is hitherto too low to make it subject of the analyses. Furthermore, it is difficult to measure duration and type (full-time/part-time) of parental leave with the Belgian data, whereas the Swedish yearly register data make it impossible to measure the exact timing of parental leave uptake.

To conclude, many countries throughout Europe exhibit low parental leave uptake by fathers (OECD 2016b). As this paper corroborates the importance of (partners') income and workplace characteristics in parental leave decisions, it provides insight into possible avenues for future policy design in Belgium and other countries with low levels of (fathers') parental leave uptake. Research of Geisler and Kreyenfeld (2018) and Patnaik (2018) on policy reforms with respect to parental leave, i.e., increasing the replacement benefits and introducing (longer) 'daddy quotas' in Germany and Quebec, for example, shows a significant increase in fathers' leave use after these reforms. Also research on the UK underlines the need for a statutory individual entitlement to well-paid leave in view of obtaining higher male parental leave uptake (Fox et al. 2009; Koslowski and Kadar-Satat 2019). Furthermore, especially higher flexibility—e.g., uptake in days or even hours, such as in Sweden—may be key to

increase fathers' parental leave use in countries with low levels of uptake by fathers as it reduces the income loss and time of absence from the workplace in case of parental leave uptake. As of 1 June 2019, changes in Belgian legislation make it possible to take full-time leave in weeks and part-time leave in months, yielding more flexibility in the uptake. However, this new form of flexibility is not a right, but an option that is only possible in agreement with the employer. Hence, the actual impact of this policy reform is yet to be addressed. It should, however, be noted that copying family policies from one country to another, with a different culture and institutional context, will not necessarily translate into similar outcomes (Neyer and Andersson 2008). The way particular welfare policies are accepted and used is highly dependent on the cultural values and ideas present (Fagnani 2002; Pfau-Effinger 2004). Nevertheless, gaining insight on the labour market determinants of father's parental leave use and how they differ by countries' normative and institutional national level contexts may provide policy makers with solid knowledge in the context of promoting and supporting more gender equal parenting roles.

**Author Contributions:** Conceptualization, L.M., J.W., A.-Z.D., E.M.; Data curation, L.M. and E.M.; Formal analysis, L.M., E.M.; Funding acquisition, J.W.; Investigation, L.M., E.M.; Methodology, L.M., E.M.; Project administration, L.M.; Resources, E.M.; Software, L.M. and E.M.; Supervision, J.W., A.-Z.D.; Visualization, L.M.; Writing—original draft, L.M.; Writing—review & editing, J.W., A.-Z.D.

**Funding:** This research was funded by the Research Foundation Flanders (FWO), grant Number G.066217N.

**Conflicts of Interest:** The authors declare no conflicts of interest.

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
