# Peer review of "Fathers’ Parental Leave Uptake in Belgium and Sweden: Self-Evident or Subject to Employment Characteristics?"

_socsci, doi:10.3390/socsci8110312_

Round 1

Reviewer 1 Report

The manuscript addresses a very relevant topic, that is, fathers' parental leave uptake in Belgium and Sweden. The topic is at the core of work-family reconciliation policies.

The manuscript is well written. The various arguments are well articulated and the sections are well organized around a very clear structure. The description of the two national contexts is very clear although it is very rich of country specific details.

Some minor comments are listed underneath.

-Hypothesis 1 is a bit too fragmented. I would state it in a more compact way and in a second step articulate it into different sub-hypotheses.

-Linear probability models are considered superior to logistic regression models, if interactions terms are included in the models. In this case, I would also specify (perhaps in a footnote) the reasons, otherwise the motivation seems to be a bit vague. Which kind of bias would be at stake using logistic regression models?

-I missed a discussion on the huge 'sample' size differences of the two countries. Given the remarkable differences, I would not give to much emphasis to the statistical significance.

-The authors did not discuss about the generalizability of the results to other countries,  and about how and to what extent they could be used to design further policy measures in (other) low fertility countries. I would advice some elaboration on this issue.

Reviewer 2 Report

I really like the article. I think it is well structured and clearly presented and I think that the subject is particularly relevant (as it is one of my main research topics).

The only aspect that I would suggest to include is some background data on parental leave uptake by fathers in Belgium as for the moment the Authors showed only those of Swedish fathers (p. 5, 192). This would help international readers to frame the article in a better way.

Moreover, I would suggest the A. to rename the section 2.1 (p. 6) because in that section they also talk about workplaces' characteristics and norms which cannot be reduced to "intra-household negotiation" and "microeconomic" choices. In my view, the gendered culture of workplaces is a meso level between couples' negotiation and the macrolevel of policies and childcare services.
